# Tumors of the Nose and Paranasal Sinuses: Promoting Factors and Molecular Mechanisms—A Systematic Review

**DOI:** 10.3390/ijms24032670

**Published:** 2023-01-31

**Authors:** Daniela Lucidi, Carla Cantaffa, Matteo Miglio, Federica Spina, Matteo Alicandri Ciufelli, Alessandro Marchioni, Daniele Marchioni

**Affiliations:** 1Department of Otolaryngology, Head and Neck Surgery, University Hospital of Modena, 41124 Modena, Italy; 2Respiratory Diseases Unit, Department of Medical and Surgical Sciences, University Hospital of Modena, 41124 Modena, Italy

**Keywords:** sinonasal squamous cell carcinoma, intestinal-type adenocarcinoma (ITAC), olfactory neuroblastoma (ONB), Biomarkers, paranasal sinus cancer, immunotherapy, targeted therapies

## Abstract

Sinonasal neoplasms are uncommon diseases, characterized by heterogeneous biological behavior, which frequently results in challenges in differential diagnosis and treatment choice. The aim of this review was to examine the pathogenesis and molecular mechanisms underlying the regulation of tumor initiation and growth, in order to better define diagnostic and therapeutic strategies as well as the prognostic impact of these rare neoplasms. A systematic review according to Preferred Reporting Items for Systematic Review and Meta-Analysis criteria was conducted between September and November 2022. The authors considered the three main histological patterns of sinonasal tumors, namely Squamous Cell Carcinoma, Intestinal-Type Adenocarcinoma, and Olfactory Neuroblastoma. In total, 246 articles were eventually included in the analysis. The genetic and epigenetic changes underlying the oncogenic process were discussed, through a qualitative synthesis of the included studies. The identification of a comprehensive model of carcinogenesis for each sinonasal cancer subtype is needed, in order to pave the way toward tailored treatment approaches and improve survival for this rare and challenging group of cancers.

## 1. Introduction

Sinonasal cancers are a heterogeneous group of tumors with an incidence of 0.83 per 100,000 individuals [1]. They account for less than 1 out of 20 of head and neck malignancies [2]. The most common entity is sinonasal squamous cell carcinoma (SNSCC), followed by intestinal-type adenocarcinoma (ITAC), and other subtypes, such as olfactory neuroblastoma (ONB), sinonasal neuroendocrine carcinoma (SNEC), sinonasal undifferentiated carcinoma (SNUC), and sinonasal melanoma [3]. Accurate diagnosis can be challenging and requires several diagnostic markers, especially for some subtypes. They typically present during the 50th to 60th decade of life, with a higher prevalence in males [4]. Occupational exposure, genetic mutations, and viral infections can be considered etiological agents in several sinonasal tumors. Overall poor survival is largely due to the biology of the different cancer types, frequent presentation at an advanced stage and limited treatment options for advanced disease, especially when surgery and radiotherapy are not a curative option anymore. Recent developments have been driven by advances in immunohistochemistry and molecular analysis. Better understanding of the pathogenesis, along with the possibility to differentiate distinct subtypes, has important implications for clinical practice, especially in the development and delivery of targeted treatment. However, due to the rarity of sinonasal cancer and its subtypes, molecular studies are still limited, and research is lagging behind other major cancers. The aim of the present review was to summarize the scientific evidence regarding molecular mechanisms behind the most common sinonasal malignant tumors.

## 2. Methods

This review was conducted in accordance with the Preferred Reporting Items for Systematic Review and Meta-Analysis (PRISMA) process to identify published articles regarding molecular pathways and carcinogenesis in malignant sinonasal tumors. Manuscripts were screened by MEDLINE database, Cochrane review, LILACS, Web of Science, and Google Scholar. Parentheses and Boolean operators (AND, OR) were applied to create conjunctions. The search was performed between September and October 2022 based on MeSH terms, as follows: [(Intestinal type adenocarcinoma OR ITAC OR squamous cell carcinoma OR chordoma OR Chondrosarcoma OR olfactory neuroblastoma) AND (nasal OR paranasal OR nose OR Sinus OR Sinusal) AND (biology OR molecular OR immunology OR biological OR oncogene OR carcinogenesis OR oncogenesis)]. The authors chose to focus the search only on selected sinonasal pathologies (namely: SNSCC, ITAC, ONB, chordoma, and chondrosarcoma), as the publications concerning other pathological entities, such as SNUC or SNEC, retrieved a small number of non-specific articles. Other entities, such as adenoid cystic carcinoma or mucosal melanoma, were excluded from the search since they were considered as arising from ectopic cells present in the nasal tissues (salivary tissue and melanocytes, respectively), whereas the present research was aimed at describing only primary tumors of nasosinusal tissues. Other exclusion criteria were: publication year < 2000; no full text available; language different from English, Italian, French, or Spanish; case reports, systematic reviews, metanalyses, and editorial letters; histological types different than the included one; anatomical subsites different than sinonasal region; articles not dealing directly with the investigated issues.

In the first screening, authors independently read the titles and abstracts of all articles performing the first selection, being as inclusive as possible. Any disagreements were resolved by consensus. In the second phase, the full articles were collected for the analysis, based on the exclusion criteria. Additional studies were manually identified from the reference lists of retrieved literature. The authors extracted data from included articles using a standardized template and collected them into a computerized database. Finally, the authors performed a qualitative synthesis of the included studies, categorized into the main pathological entities.

## 3. Results and Discussion

In total, our search yielded 1450 articles. A further manual check of the references included in the articles was performed, adding 7 articles. We excluded 208 articles for not dealing directly with the investigated issue, 364 articles for publication year < 2000, 14 records for the full text not being available, 63 for being written in a language different than the included ones, and 15 for being about veterinary medicine. Finally, 793 full text articles were assessed for eligibility and 39 were excluded for describing different histological entities, 281 for anatomical subsites different than sinonasal region, and 227 for being case reports, systematic reviews, metanalyses, or editorial letters. Finally, 246 articles were included in the qualitative analysis. The details of the systematic search are shown in Figure 1. Since the large majority of the retrieved articles examined SNSCC, ITAC, and ONB, the discussion was focused on these three pathological entities.

### 3.1. Sinonasal Squamous Cell Carcinoma

As known, SNSCC may arise from benign lesions with malignant transformation potential. Even though a rare event, SNSCC progression from inverted papilloma (IP) provides the perfect model to understand molecular changes associated with SNSCC pathogenesis. Results regarding the most commonly studied immunohistochemical markers of SNSCC are summarized in Table 1. Increased cell proliferation unopposed by programmed cell death, as seen in many other malignancies, is one proposed mechanism behind IP transformation. All studies on the subject agree on increased expression of Ki-67, a known marker of cell proliferation, during this process [5,6,7,8,9,10,11,12,13,14]. However, due to heterogeneity in the tissues analyzed and the interpretation of immunostaining, these results cannot be summarized together. Interestingly, primary SNSCC seems to have a lower Ki-67 expression compared to IP-associated SNSCC, suggesting a different biology of the two lesions [6].

Ciclin-dependent kinase (CDK) complexes, which promote transition between different phases of the cell cycle, and CDK inhibitors (CDKI) have also been found to be deregulated in IP with malignant transformation potential and in SNSCC.

For instance, expression of the CDKI p27 was observed to be lower in both primary SNSCC and IP-derived SNSCC with respect to IP alone [5,10,11].

Data on p21, another CDKI, are, however, not as convincing, with two studies reporting a progressive increase in p21 immunostaining during IP progression [15,16]. Increased expression of a pro-apoptotic molecule during a carcinogenic process may sound counterintuitive; however, it should be noted that it probably reflects the accumulation of the mutated form of p21, which, unlike its wild-type counterpart, is unable to block the cell cycle.

In a study by Katori and colleagues, progressively higher p21 staining during IP progression was accompanied by a similar increase in p53 expression. This finding led the authors to hypothesize that, with p21 being a downstream effector of p53 anti-proliferative activity, wild type 53-dependent p21 overexpression occurs in IP [16].

Actually, a few other studies show consistent p53 overexpression in SNSCC, both primary and IP-derived, compared to IP alone [5,15]. However, Yoon and colleagues, who also reported an association between p53 expression and grade of dysplasia in IP, underlined that staining for p53 detects the presence of its mutant isoform, because it has a significantly longer half-life [17]. If this were true, the high p21 expression found in the previously cited studies may have to be attributed to a p53-independent mechanism.

In favor of this hypothesis, next-generation sequencing studies have reported loss of function TP53 mutation as a frequent event in IP-derived SNSCC [18,19].

In light of this evidence, it can be hypothesized that p53 and p21 mutations may be two independent mechanisms which both result in anti-apoptotic activity in cancer cells. Whether these two carcinogenic pathways may coexist in the same patient is a matter of debate. In fact, opposite to Katori et al., Bandoh et al. observed p53 mutation to be associated with loss of p21 expression [9].

Going along the line of cell cycle regulatory molecules, expression of cyclooxygenase-2 (COX-2), an inflammation-inducible factor with proliferative and anti-apoptotic activity, was found to be progressively higher as IP progresses to SNSCC, and it also seems to be higher in primary SNSCC compared to IP alone. This molecule, therefore, has been implicated in inflammatory-driven carcinogenesis in SNSCC. Higher COX-2 expression may be due to p53 mutation, as wild-type p53 normally suppresses COX-2 transcription [17,20].

Other than cell cycle regulation, COX-2 has been attributed a role in neoangiogenesis and cell invasion as additional tumorigenic properties.

Neoangiogenesis, as a matter of fact, despite being a well-known tissue step of malignant transformation in various districts, has been poorly investigated in the setting of SNSCC. Overall, it seems that IP-derived SNSCC is characterized by aberrant angiogenesis, as demonstrated by higher vascular density and increased Vascular Endothelial Growth Factor (VEGF) expression in IP-derived SNSCC than in IP alone. Not surprisingly, higher levels of maximum vessel density were found in tumors with neck and/or distant metastasis and in patients with poorer prognosis. Interestingly, increased VEGF expression was found to be a powerful prognostic marker in SNSCC but not in other tumors of the sinonasal tract [13,21,22].

A few studies on the increased expression of metalloproteinases (MMP) have been published regarding how IP cells acquire invasion capabilities [23]. In one interesting study, Shin et al. observed a high frequency of 19p13.3 gain in IP and IP-derived SNSCC. This region includes 30 protein-coding genes, one of which is BSG, encoding for the extracellular matrix MMP inducer (EMMPRIN)/CD147. This region also contains seven genes related to inflammatory cell function, including *PRTN3, GZMM, PRSS57*, and *ELA2,* which encode proteases secreted by inflammatory cells, and *AZU1*, which encodes a chemotactic glycoprotein stored in neutrophils [24]. This finding, coupled with the previously cited reports on COX-2, suggests that there may be a role for inflammatory-driven carcinogenesis in SNSCC as there is in other malignancies. Interestingly, prevalence of granulocytes and mononuclear phagocytes as opposed to T cells in the tumor microenvironment of SNSCC was found to be associated with recurrent disease [25], while tumor infiltration by CD8+ T cells seems to be associated with better progression-free survival rates [26]. However, a study by García-Marín et al. observed tumor-infiltrating CD8+ T cells to predict worse survival, but suggested that CD8+ T cells-infiltrated SNSCC may benefit from immunotherapy [27]. While the prognostic contribution of tumor-infiltrating lymphocytes is unclear, there may still be a role for immunotherapy in SNSCC. In fact, increased expression of PD-L1, a known predictor of immunotherapy efficacy, characterizes a subset of SNSCC [26,27,28,29].

Apart from COX-2 and CDK inhibitors, other known regulators of programmed cell death have also been investigated in the process of SNSCC pathogenesis, mainly the antiapoptotic protein Bcl-2 and the proapoptotic protein Bax, though with contrasting results [6,8,17]. Interestingly, Katori et al. demonstrated a significant increase in the apoptotic index (AI) from controls to IP with dysplasia, IP with carcinoma, and invasive SNSCC and, more importantly, they observed a significant increase in the AI in IP with mild and moderate dysplasia compared with IP with carcinoma and invasive SNSCC [6]. Similarly, in a study by Fan et al., the AI was found to be significantly lower in primary SNSCC than in IP [8]. To sum up, apoptosis is likely to increase in IP with respect to normal controls, as a result of increased proliferation; however, during the progression of IP to SCC, despite proliferation increases, the rate of apoptosis does not rise accordingly, resulting in a net prevalence of proliferation over cell death.

**Table 1 ijms-24-02670-t001:** Summary of immunohistochemical markers of inverted papilloma (IP) and SNSCC (Sinonasal Squamous Cell Carcinoma). CIS (Carcinoma in situ); NKCa (Non-Keratinizing Carcinoma); KSCC (Keratinizing SCC).

Marker	Article (Authors, Publication Year)	Unit of Measurement	IP	IP with SCC	Primary SCC	Unspecified SCC	Statistical Difference
Ki-67	Oncel S et al., 2011 [5]	% of positive cases	10.77		35.33		*p* < 0.05
	Katori H et al., 2007 [6]	Mean % of positive area	8.5 (with mild dysplasia) 9 (with moderate dysplasia) 21.12 (with severe dysplasia)	26.5	18.8		A significant increase was observed in IP with severe dysplasia, IP with SCC, and primary SCC compared with IP with mild and moderate dysplasia (*p* < 0.05).
	Katori H et al., 2005 [7]	Mean % of positive area	NA	NA	NA		Significant increase was observed in IP with severe dysplasia, IP with SCC, and primary SCC compared to IP with mild and moderate dysplasia (*p* < 0.05).
	Fan G et al., 2006 [8]	Ki-67 index	22.3 (without dysplasia) 19.2 (IP portion of IP with dysplasia) 22.5 (dysplasia portion of IP with dysplasia)	21 (IP portion) 47.4 (CIS portion) 61 (SCC portion)	60.8		Ki-67 index was higher in primary SCC compared with IP (*p* = 0.0002), in the portion of in situ SCC than in the IP portion (*p* = 0.0048), and in the portion of invasive SCC than in the IP portion (*p* = 0.0019).
	Bandoh N et al., 2005 [9]	% of positive cases				24	/
	Schwerer MJ et al., 2002 [10]		13.6 (columnar) 20.1 (transitional) 16.9 (squamous) 50.2 (IP with dysplasia)	71 (CIS)	38.4 (NKCa) 30 (KSCC)		Transitional and squamous epithelium showed significantly higher Ki-67 immunopositivity compared with columnar epithelium in IP (*p* < 0.05). Significantly higher expression rates of Ki-67 were present in dysplastic compared with non-dysplastic IP (*p* < 0.05). The difference between CIS and dysplastic IP and between SCC and CIS was not statistically significant. The expression rates of Ki-67 were comparable between NKCa and KSCC.
	Tsou Y et al., 2014 [11]		NA	NA			Elevated Ki-67 was found in IP with SCC compared with IPs alone (*p* = 0.001).
	Hakim SA et al., 2021 [12]		20.5	31.43			Yes (*p* value not available).
	Valente G et al., 2006 [13]	% of positive cases				50	/
	El-Mofty SK et al., 2005 [14]	% of positive cases				100 (KSCC) 100 (NKCa)	/
p21	Oncel S et al., 2011 [5]	% of positive cases	NA		NA		No
	Kim S et al., 2011 [15]	% of positive cases	12.5		77.8		*p* < 0.05
	Bandoh N et al., 2005 [9]	% of positive cases				9	/
	Katori H et al., 2006 [16]	% of positive cases	14.3 (with mild dysplasia) 37.5 (with moderate dysplasia) 57.1 (with severe dysplasia)	71.4	50		Significant increased staining of p21 was observed in IP with severe dysplasia and IP with carcinoma compared with IP with slight dysplasia (*p* value not available).
	Kakizaki T et al., 2017 [30]		NA	NA	NA		No significant differences were observed in p21 expression between non-dysplastic and dysplastic IPs, or between SCCs with IPs and SCCs without IPs.
	Tsou Y et al., 2014 [11]		NA	NA			No
p27	Oncel S et al., 2011 [5]	% of positive cases	NA		NA		There was a marked decrease in the expression of p27Kip1 in the SCC group compared to the IP group, but this was not statistically significant.
	Bandoh N et al., 2005 [9]	% of positive cases				12	/
	Schwerer MJ et al., 2002 [10]		63.1 (without dysplasia) 68 (with dysplasia)	39.4 (CIS) 25.1 (NKCa) 42.1 (KSCC)			Significantly reduced p27Kip1 expression was found in surface cells in dysplastic compared with non-dysplastic IP, as well as in the total number of cells in carcinoma in situ compared with dysplastic inverted papillomas (*p* < 0.05).
	Tsou Y et al., 2014 [11]		NA	NA			Elevated p27 was found in IP with SCC compared with IPs alone (*p* = 0.001).
p63	Oncel S et al., 2011 [5]	% of positive cases	22.7		77.8		*p* = 0.019
	Kim S et al., 2011 [15]	% of positive cases	100		100		No
cox-2	Yoon B et al., 2013 [17]	NA					Data do not fit within the categories of this table.
	Lee G et al., 2012 [20]		5.4 (without dysplasia) 0 (with dysplasia)	38.9	41.7		The percentage of positive tumors was significantly higher in IPs with SCC and primary SCCs compared with benign IPs (*p* = 0.000) and in primary SCC compared with IP with dysplasia (*p* = 0.44).
bcl-2	Fan G et al., 2006 [8]	Mean expression rate (%)	2.2 (without dysplasia) 1.2 (IP portion of IP with dysplasia) 2 (dysplasia portion of IP with dysplasia)	2.1 (IP portion) 3.6 (CIS portion) 4.3 (SCC portion)	6.4		No
	Yoon B et al., 2013 [17]	NA					Data do not fit within the categories of this table.
	Katori H et al., 2007 [6]		1.8 (with mild dysplasia) 3.1 (with moderate dysplasia) 5.2 (with severe dysplasia)	5.3	6.2		A significant increase was observed in IP with severe dysplasia and carcinoma and invasive SCC compared with IP with mild dysplasia (*p* < 0.05).
bcl-xL	Bandoh N et al., 2005 [9]	% of positive cases				47	/
	Bandoh N et al., 2003 [22]	% of positive cases				47	/
	Bandoh N et al., 2001 [31]	% of positive cases				47	/
bax	Takahashi Y et al., 2013 [32]	% of positive cases				18.8	*p* = 0.0186 (with respect to normal tissue).
	Bandoh N et al., 2005 [9]	% of positive cases				57	/
	Bandoh N et al., 2003 [22]	% of positive cases				57	/
	Bandoh N et al., 2001 [31]	% of positive cases				57	/
	Fan G et al., 2006 [8]	Mean expression rate (%)	85.3 (without dysplasia) 84 (IP portion of IP with dysplasia) 88.5 (dysplasia portion of IP with dysplasia)	86.5 (IP portion) 83.2 (CIS portion) 77.2 (SCC portion)	59.5		Bax expression was significantly higher in primary SCC than in IP without dysplasia (*p* < 0.05).
	Yoon B et al., 2013 [17]	NA					Data do not fit within the categories of this table.
p53	Takahashi Y et al., 2013 [32]	% of positive cases				39.1	*p* = 0.0001 (with respect to normal tissue).
	Bandoh N et al., 2005 [9]	% of positive cases				56	/
	Bandoh N et al., 2003 [22]	% of positive cases				56	/
	Bandoh N et al., 2001 [31]	% of positive cases				56	/
	Tsou Y et al., 2014 [11]	NA	NA	NA			No
	Oncel S et al., 2011 [5]	% of positive cases	0		33.3		*p* = 0.015
	Kim S et al., 2011 [15]	% of positive cases	0	44.4			*p* < 0.05
	Fan G et al., 2006 [8]	Mean expression rate (%)	2.4 (without dysplasia) 8.1 (IP portion of IP with dysplasia) 10.4 (dysplasia portion of IP with dysplasia)	9.3 (IP portion) 11.5 (CIS portion) 24.6 (SCC portion)	37.5		p53 expression was higher in primary SCC with respect to IP (*p* = 0.016). There also was a trend for p53 to accumulate in IP with dysplasia and IP with SCC.
	Yoon B et al., 2013 [17]	NA					Data do not fit within the categories of this table.
	El-Mofty SK et al., 2005 [14]	% of positive cases				27.8 (KSCC) 37.7 (NKCa)	/
	Scheel A et al., 2015 [33]	% of positive cases	51.9				/
	Katori H et al., 2005 [7]	Mean % of positive area	NA	NA	NA		Significant increase was observed in IP with severe dysplasia, IP with carcinoma, and invasive SCC compared to IP with mild and moderate dysplasia (*p* value not available).
	Katori H et al., 2006 [16]	% of positive cases	14.3 (with mild dysplasia) 25 (with moderate dysplasia) 57.1 (with severe dysplasia)	57.1	66.7		A significant increase was observed in IP with severe dysplasia, IP with SCC, and primary SCC compared with IP with mild dysplasia (*p* < 0.05).
HER-2/ErbB2	Takahashi Y et al., 2013 [32]	% of positive cases				2.9	No (with respect to normal tissue).
	Li et al., 2019 [34]	Mean mRNA expression level and mean protein expression level	NA	NA			mRNA and protein expression levels were higher in IP than in normal nasal mucosa (*p* < 0.01). Protein expression levels were higher in the IP and IP with SCC than in normal nasal mucosa (*p* < 0.01). Protein expression levels were significantly higher in IP with SCC than IP (*p* < 0.01).
	López F et al., 2011 [35]	% of positive cases				7	/
EGFR	Takahashi Y et al., 2013 [32]	% of positive cases				82.1	*p* < 0.0001 (with respect to normal tissue).
	Katori H et al., 2005 [7]	Mean % of positive area	NA	NA	NA		Significant increase was observed in IP with severe dysplasia, IP with SCC, and primary SCC compared to IP with mild and moderate dysplasia (*p* < 0.05).
	López F et al., 2011 [35]	% of positive cases				39	/
	Hongo T et al., 2021 [26]	% of positive cases				77.1	/
	Jiromaru R et al., 2019 [36]	% of positive cases				77.2	/
	Chao J et al., 2008 [37]	% of positive cases	80	100 (synchronous) 90 (metachronous)			NA
	Scheel A et al., 2015 [33]	% of positive cases	61				/
VEGF	Takahashi Y et al., 2013 [32]	% of positive cases				40.3	*p* = 0.0073 (with respect to normal tissue).
	Valente G et al., 2006 [13]	% of positive cases				41.1	/
	Yu H et al., 2013 [21]	Relative protein level	NA	NA			Protein expression was increased in IP and IP with SCC with respect to normal nasal mucosa (*p* < 0.01), and in IP with SCC with respect to IP alone (*p* < 0.05).
	Bandoh N et al., 2003 [22]	% of positive cases				50	/
TGF-a	Katori H et al., 2005 [7]	Mean % of positive area	NA	NA	NA		A significant increase was observed in IP with SCC and primary SCC compared to IP with mild and moderate dysplasia (*p* < 0.05).
	Li et al., 2019 [34]	Mean mRNA expression level	NA				TGF-a mRNA expression levels were significantly higher in the IP than in the normal nasal mucosa (*p* < 0.01).
Cyclin-D1	Takahashi Y et al., 2013 [32]	% of positive cases				57.6	*p* < 0.0001 (with respect to normal tissue).
	Scheel A et al., 2015 [33]	Mean % of positive area	79.7				/
Fas	Bandoh N et al., 2005 [9]	% of positive cases				29	/
	Bandoh N et al., 2003 [22]	% of positive cases				29	/
	Bandoh N et al., 2001 [31]	% of positive cases				29	/
	Fan G et al., 2006 [8]	Mean expression rate (%)	78.2 (without dysplasia) 81.2 (IP portion of IP with dysplasia) 79.5 (dysplasia portion of IP with dysplasia)	83.8 (IP portion) 84 (CIS portion) 81.1 (SCC portion)	75.5		No

Finally, research has heavily focused on HPV infection as a potential driver of IP malignant transformation, this agent being well known for its pathogenic role in oropharyngeal lesions.

However, the prevalence of HPV infection in IP and SNSCC, either IP-derived or primary, greatly varies among studies, as does the HPV genotype putatively involved (low-risk vs. high-risk). Additionally, there is no consensus on which is the best method to identify HPV infection in these lesions, with different authors alternatively using in situ hybridization, PCR, and/or p16 immunostaining (Table 2). In fact, p16 has been reported to have a low sensitivity in detecting HPV infection. This may be due to low levels of HPV transcriptional activity in tumor cells, which poses a question mark on HPV role in SNSCC pathogenesis [38,39].

Some authors reported a higher incidence of HPV infection in pre-malignant lesions compared with SNSCC, suggesting that HPV infection may be an early event in IP malignant transformation, which nonetheless goes on even when the infection has been cleared by the host immune system [23]. In support of this hypothesis, Paehler vor der Holte et al. showed that in three of four analyzed cases of IP-derived SCC, the benign part of the tumor was HPV-positive, but no virus could be detected in the SCC portion. Furthermore, one patient initially presented with benign papilloma and returned after years with recurrent SCC ex-papilloma. In this patient, HPV DNA was detected in the primary tumor but not later in the SCC [40].

On the other hand, in a study by Udager et al. on 13 matched IP and IP-derived SCC samples, HPV status was the same in all pairs [38]. Additionally, Sahnane et al. observed high risk HPV in a small percentage of SNSCC (both primary and IP derived) and in no case of IP alone [39]. These results not only go against the hypothesis of IP-derived SNSCC growing independent of HPV infection, but also show an overall low prevalence of HPV infection in both IP and IP-derived or primary SNSCC, which has also been reported by other authors [41,42,43].

Interestingly, HPV, in particular the 16 subtypes, has been shown to be more frequently associated with non-keratinizing SNSCC [14,36,44].

Reviewing the literature on HPV, we came across one interesting study by Udager and colleagues suggesting HPV infection as one of two mutually exclusive pathogenic mechanisms in IP malignant transformation, the other one being an activating mutation of the EGFR gene [38]. Similar findings were shown by Hongo et al. [26,45] and Jiromaru et al. [36]. On the opposite, Katori et al. [7] and Scheel et al. [33] observed a significant increase in EGFR expression in HPV positive IP compared to HPV negative IP. Interestingly, HPV infection seems to be associated with a higher risk of progression to malignancy in IP, while EGFR mutation, irrespectively of HPV status, may be protective for the outcome [33,38,39,46].

On the opposite, in overt SNSCC, HPV positivity seems to be a predictor of longer disease free and overall survival [36,44,45,47,48], as it is in oropharyngeal tumors, while EGFR protein overexpression seems to be associated with significantly shorter disease-free survival and overall survival [32,45,49,50].

The role of another known viral carcinogen, Epstein–Barr Virus (EBV), has also been investigated as a potential etiologic agent in SNSCC pathogenesis. However, in contrast to the conspicuous literature on HPV, there were only two studies on EBV in our literature search. Interestingly, both studies observed a significant association between EBV infection and lymph node metastasis [51,52]. This finding not only sheds light on the pathogenesis of SNSCC metastatic spread, but may also be exploited to personalize the therapeutic strategy, reserving cN0 lymph node dissection to EBV-positive cases.

**Table 2 ijms-24-02670-t002:** HPV expression in sinonasal squamous cell carcinoma and precursors lesions. IP: inverted papilloma, ISH: in situ hybridization, PCR: polymerase chain reaction; LR-HPV: low risk HPV, HR-HPV: high risk HPV, ICH: immunohistochemistry, KSCC: keratinizing sinonasal SCC, NKCa: non-keratinizing sinonasal SCC.

Article (Author, Publication Year)	HPV+ Cells (%)					Technique
	IP without Dysplasia	IP with Dysplasia	IP with SCC	Primary SCC	SCC (Unspecified)	
Katori H et al., 2006 [16]		mild: 16.7 (HPV 6/11); 0 (HPV 16/18) moderate: 33.3 (HPV 6/11); 33.3 (HPV 16/18) severe: 71 (HPV 6/11); 57 (HPV 16/18)	57 (HPV 6/11) 43 (HPV 16/18)	25 (HPV 6/11) 17 (HPV 16/18)		HPV DNA ISH
Katori H et al., 2006 [23]		severe: 63 (HPV 6/11); 50 (HPV 16/18)	57 (HPV 6/11) 43 (HPV 16/18)	25 (HPV 6/11) 17 (HPV 16/18)		HPV DNA ISH
Hoffman M et al., 2005 [42]	11.5 (HPV 6+11)				20 (HPV 16)	HPV DNA PCR
McKay SP et al., 2005 [53]	9.1 (HPV 11)		66.7 (HPV18)			HPV DNA PCR
Paehler vor der Holte A et al. 2020 [40]	22.5 (LR-HPV) 15.5 (HR-HPV)		47.6 (HR-HPV)			HPV DNA PCR
Katori H et al., 2005 [7]		severe: 57.1 (HPV 6/11); 42.9 (HPV 16/18)	66.7 (HPV 6/11) 50 (HPV 16/18)	25 (HPV 6/11) 17 (HPV 16/18)		HPV DNA ISH
Chatzipantelis P et al., 2020 [54]	78.6				66.7	p16 IHC
Sahnane N et al., 2019 [39]	0 (HR-HPV) 24 (LR-HPV)		13 (HR-HPV) 0 (LR-HPV)	8 (HR-HPV) 0 (LR-HPV)		HPV DNA ISH
Mohajeri S et al., 2018 [41]	11.4	severe: 20 (HPV 6/56)	0			HPV DNA PCR
Udager AM et al., 2017 [38]	10.3 (HPV 6/11)		22.7 (18.2 LR, 4.5 HR)	35.7 (28.6 HR, 7.1 unknown)		HPV DNA PCR
Rooper LM et al., 2016 [43]	0		6	43		p16 IHC
	0		0	29		HR-HPV RNA ISH
Scheel A et al., 2015 [33]	12.2 (LR-HPV)					HPV DNA PCR
Stoddard Jr DG et al., 2015 [55]	12.5		0			p16 IHC, LR-HPV DNA PCR, mRNA ISH
Larque AB et al., 2014 [48]					20 (HR-HPV)	HPV DNA PCR, HR-HPV DNA ISH, p16 IHC
El-Mofty SK et al., 2005 [14]					KSCC: 19 (HPV-16) NKCa: 50 (HPV-16)	HPV DNA PCR
					4.8 (KSCC) 62.5 (NKCa)	p16 IHC
Tendron A et al., 2022 [47]					18.6	p16 IHC
Hongo T et al., 2021 [26]					11.5	p16 IHC
					6.1 (HR-HPV)	HPV mRNA ISH
Nishikawa D et al., 2021 [49]					8	HPV DNA PCR
Laco et al., 2015 [44]					32.7	p16 IHC
					19	HPV DNA PCR
					24.5	HPV DNA ISH
Vietía D et al., 2014 [56]					8.3	Reverse hybridization
Jiromaru et al. 2019 [36]					14.9	p16 IHC
					8.9	HPV RNA ISH
Takahashi Y et al., 2013 [32]					9.4	HPV DNA ISH
					17.9	p16 IHC
Doescher J et al., 2015 [51]					20.5	HPV DNA ISH
					29.5	p16 IHC

### 3.2. Intestinal Type Adenocarcinoma

A tight link exists between exposure to wood dust (WD) or to leather tannin and development of ITAC, but little is still known about the molecular mechanisms at work in the process [57,58,59,60,61,62]. In 1995, the International Agency for Research on Cancer (IARC) classified WD as a carcinogen in humans [63]. Demers et al. [64] quantified risk related to occupational WD exposure: this risk increases with exposure intensity (mg/m^3^ of atmospheric dust) and with exposure duration. ITAC, however, may develop even after short exposure: 9.6% of patients had less than 5 years of exposure and in 13.2% of patients, the interval between exposure and diagnosis exceeded 40 years [57,59]. These data suggest the necessity of lifetime monitoring in professionally exposed patients.

Gallet et al. [65] found transcriptome modifications in the contralateral nasal fossa of WD-exposed ITAC patients; no such modifications were found in WD-exposed healthy patients, thus leading to the hypothesis of a specific sensitivity to WD. No genetic risk factors have been identified yet [57]; to our knowledge, only a study associated polymorphism rates in codon 461 of CYP1A1 with ITAC (23.3% vs. 7.6% in controls, *p* = 0.046) [66].

Several studies suggested that ITAC develops by chronic inflammation caused by prolonged exposure and irritation by WD particles, stimulating cellular turnover [67,68]. It has been shown that the true risk factor is the actual exposure to WD particles and not the possible exposure to chemical products used in the industry [69]. Holmila et al. [61] found an association between expression of COX2, an enzyme largely responsible for inflammation, as previously mentioned, and ITAC, which was stronger in WD-exposed ITAC patients, supporting a role of WD exposure in eliciting inflammatory response and consequent ITAC development. However, other authors believe that chronic inflammation would not explain the development of ITAC; in fact, factors associated with chronic inflammation, such as smoking, seem to have a role in SCC development, but not in ITAC. Moreover, the interval between exposure to WD and the diagnosis of ITAC is rather long even in patients with less than 5 years of exposure.

ITAC owes its name to the pathological similarity with the colorectal cancer (CRC). The key-gene in the regulation of intestinal differentiation is CDX2, that is, the pre-requisite of intestinal metaplasia, of which the overexpression within ITAC cells is almost systematic [65,70,71,72,73,74,75]. Some authors suggested that the developing process of ITAC is similar to that in colic adenocarcinoma because of its histopathological resemblance. In fact, ITACs commonly express CK20, typical in the intestinal cells, the intestinal cytoskeletal protein villin, and CDX2; they could also express CK7, which is a cytokeratin of the normal respiratory epithelium, SATB2, expressed in the epithelium of the lower gastrointestinal tract and in a large majority of colorectal adenocarcinomas, and MUC2 and MUC5, which are markers expressed in the digestive tract [70,72,73,74,76,77,78,79,80,81]. This suggests that the development of ITAC is preceded by intestinal metaplasia (IM) [70,72]. Indeed, areas of IM have been observed in proximity to ITAC and have been considered as preneoplastic lesions [71,72,82,83].

However, differences between ITAC and CRC are not negligible, and most authors sustain that ITAC and colorectal adenocarcinoma have different genetic pathways of development and progression and different gatekeeper events [68,84,85]. In fact, the WNT-APC-β-catenin (CTNNB1 gene) pathway and EGFR/ras/raf pathway mutations are less frequent in ITAC than in CRC [84,85,86,87,88,89,90,91,92], thus leading to the hypothesis that the gatekeeper mutation of ITAC development is not APC, which, on the contrary, is the key gene in the development of the most frequent subtype of CRC.

An interesting analogy between ITAC carcinogenesis and adenocarcinoma developing from Barrett’s esophagus was described [57]. Both nasal respiratory mucosa and esophageal mucosa derive from the primitive digestive tube. During embryogenesis, the distal digestive tube becomes differentiated by overexpression of CDX2, which, in contrast, is inactivated in the proximal tube [93]. The most attractive hypothesis to explain the metaplastic switch is that specific agents (WD for nasal adenocarcinoma and bile acids for esophageal adenocarcinoma) may release the inhibition exerted during embryogenesis by CDX2 promoter methylation, thus leading to progression toward metaplasia [57,71]. A hypothesis is that TP53 tumor suppressor gene mutation plays an important role in IM progression to ITAC, as it is in Barrett’s esophagus progression to esophageal carcinoma [94]. p53 overexpression, which is considered a reliable indicator of TP53 mutations, and TP53 mutations themselves are, in fact, not rare in ITAC [60,61,71,80,87,88,90,95,96,97,98] and seem to be related to worse grading [98]. In a CDX2-transgenic mouse model, CDX2 overexpression was associated with the appearance of foci of IM in gastric mucosa; the IM lesions frequently progressed into cancer and this process was shorter when CDX2-transgenic mice were crossed with p53-deficient mice [99]. Furthermore, Franchi et al. [71] reported that sinonasal IM associated with ITAC frequently shows overexpression of p53 and may harbor the same TP53 gene mutation present in the adjacent ITAC. Similar observations have been made in esophageal and gastric tumorigenesis [94,100,101].

TP53 mutations and p53 overexpression seem to be related to WD exposure and have been identified not only in metaplastic foci, but even in normal epithelial mucosa and stromal glands of woodworkers [60,61,88,96,102]; consequently, this occupational risk could be linked both to the molecular development of IM and to its progression to ITAC. Some authors have associated TP53 mutations and related progression to ITAC with chronic inflammatory conditions generated by WD exposure, via COX2 overexpression [61] or via free radicals, especially RNS [96]. In CRC, in fact, a strong relationship was established between NOS2 activity and TP53 mutations [103]. TP53 mutations (which can lead to loss of heterozygosity and consequent loss of its tumor suppressor role) do not entirely explain the development of ITAC, as they are found in 39% to 86% of cases [85,89,90,91,104]; further studies are needed to fully explore p53 role in ITAC carcinogenesis.

### 3.3. Olfactory Neuroblastoma

ONB is characterized by a heterogenous biological behavior. A few recurrent genomic aberrations or somatic mutations have been consistently reported by some authors [105,106], which may pave a way for new tailored treatment approaches. ONBs are characterized by chromosomal instability with highly complex copy number changes. Guled et al. confirmed some chromosomal gains and loss and reported some novel aberrations, including gains at 5q34-q35, 6p12.3, 7q11.22-q21.11, 9p13.3, 10p12.32, 12q23.1-q24.31, 13q, 20p/q, and Xp/q, and losses at 15q11.2-q24.1, 15q13.1, 19q12-q13, 22q11.1-q11.21, 22q11.23, and 22q12.1 [107]. These regions are also frequently associated with tumorigenesis in carcinomas and other histologies. Other aberrations were previously reported in the literature: the most frequent were located on chromosome arms 2q, 6q, 21q, and 22q [108,109].

Weiss et al. identified somatic alterations that might be drivers of tumorigenesis and/or metastatic progression. They found specific mutations correlating to metastatic setting, such as KDR, MYC, SIN3B, and NLRC4. On the other hand, mutations in TP53, TAOK2, and MAP4K2 were also present in primary ONB samples. However, local aggressive behavior and a more frequent recurrence rate seem to be associated with wild-type TP53 hyper-expression [110]. Similarly, Lazo de la Vega et al. confirmed these results, demonstrating that TP53 mutations are not likely to be associated with metastatic disease, while high rates of recurrence correlate with multiple copy-number chromosomal aberrations and focal amplification including CCND1 and FGFR3, which is specifically nominated as a recurrent oncogenic driver in a subset of ONBs [106]. The FGFR3 amplification leads to an FGFR3 overexpression, thus representing a potential therapeutic target with tyrosine kinase inhibitors. Conversely, the analysis of FGFR1 amplification displays no genetic aberrations in the ONB group [111].

Micheloni and colleagues performed a laboratory analysis in order to investigate the expression of OTX1 and OTX2 genes, due to the fact that these genes are expressed during embryonic morphogenesis and during the development of olfactory epithelium in adult organisms, and furthermore, gain or loss mutations in these loci could promote tumorigenesis. In ONB, they found only OTX2 gene expression, while OTX1 was downregulated. Accordingly, the activation/inactivation of OTX factors represents a significant pathogenetic event and may suggest that OTX2 could be a useful molecular marker for the diagnosis of ONB and a potential therapeutic target [112,113].

A comprehensive molecular profiling of advanced/metastatic ONB was performed by Topcagic et al. and revealed an upregulation of several genes implied in carcinogenesis, including CD24, SCG2, and IGFBP-2, but also a downregulation among ABCA8 genes, correlating with high-grade tumors, and Growth Hormone Receptor (GHR) genes [114].

Another recent comprehensive genomic profiling identified potential therapeutic targets in the mTOR, CDK, and growth factor signaling pathways [115], while PD-L1 expression in ONB samples was found to be poor, which translates into a lower chance of response to anti-PD-1/PD-L1 drugs, as reported in few studies [114,116]. A prediction of a poor response to these treatments was also underlined by some studies concerning tumor mutational burden (TMB), a biomarker that is a surrogate for the total number of genetic mutations in a tumor and predicts tumor response to immune checkpoint inhibitors. Friedman et al. showed low TMB measurements in ONB, which means a limited utility of immunotherapy treatment, but it is a novel biomarker that contributes in the classification of all cancers [117].

Molecular studies, using a multi-omic approach, were shown to be relevant in creating a subtype classification of ONB as basal or neural, both of which have distinct pathological, transcriptomic, proteomic, and immune features. Gay et al. showed that the basal subtype is characterized by mutations, especially Isocitrate Dehydrogenase 2 (IDH2) R172 [115]. This mutation underlies global epigenomic divergence in ONB with DNA hypermethylation, and consequently, may possibly lead to failure of basal cell differentiation in neuronal lineage [118]. Some authors demonstrated that the mutational status of IDH2, as well as DNA methylation patterns, could aid in the classification of ONB. They proposed a diagnostic algorithm for ONBs, divided them into four groups based on DNA methylation profiling, and in their histological analysis defined the novel entity of “Sinonasal tumors with IDH2 mutations”, which are neoplasm seem to be more similar to IDH2 mutated SNUC [119,120,121]. BUB1, a mitotic checkpoint serine/threonine kinase, is more expressed in the basal subgroup of ONBs; moreover, it has been associated with aggressive disease and poor survival [122].

Other potential druggable pathways are angiogenesis and EMT, which were significantly augmented in patients experiencing recurrence [122]. Targeting angiogenesis by blocking one or multiple axes (i.e., FGF, HGF, and VEGF) could be evaluated in the future as an alternative ONB treatment strategy [105]. EMT correlates with a worse prognosis, due to some factors that induce tumoral immune escape pathways, such as the TGF-β pathway, which is upregulated in patients with poor disease-free survival (DFS) [122].

Czapiewski et al. evaluated a potential role in ONB of some transcription factors that are frequently upregulated in tumors showing neuroendocrine differentiation. PAX5 expression was not frequent in ONBs but showed a potential correlation with an aggressive clinical course of the disease. TTF1 positiveness does not exclude the diagnosis of ONB, although only a small percentage of cells show its expression [123].

The Sonic Hedgehog (Shh) signaling pathway has a role in the regulation of tumor cell proliferation in several human neoplasms, such as the pancreas, prostate, and skin carcinoma. Aberrant Shh pathway seems to be also involved in the pathogenesis of ONB and the expression levels of Patched1, Gli1, and Gli2 appear to be correlated with its pathological degrees: Kadish stage C or Hyams grade III/IV are characterized by lower levels of membrane-associated Patched 1 and higher levels of nuclear Gli1 expression when compared with lower degrees of ONB. The blockage of Shh signaling inhibits the proliferation of ONB cells and induced ONB cell cycle arrest and apoptosis, controlling the growth of the tumor. Therefore, the Shh pathway has a crucial role in regulating the development and progression of ONB in humans, accordingly to Mao et al. [124].

Recently, Czapiewski’s group demonstrated a high prevalence of Somatostatin Receptor 2 (SSTR2) expression in ONB, which was not seen in other sinonasal histologies [125]. Data were also confirmed by Cracolici et al., who observed the overexpression of SSTR2 regardless of grade or stage [126]. The INSICA-Network suggested SSTR2 use in enabling the detection of recurrent disease and metastases, as well as its role in the implementation of peptide receptor radionuclide therapy (PRRT) as treatment [127].

## 4. Conclusions

Pathogenesis of the three most common sinonasal malignancies is currently ill defined, largely due to the rarity of these entities. A better understanding of pathogenic mechanisms behind sinonasal malignancies may pave the way towards tailored treatment approaches, and therefore, improve their prognosis. Regarding SNSCC, it seems that an imbalance between increased cell proliferation and programmed cell death drives malignant transformation of precursor lesions. Furthermore, HPV infection apparently plays a role in SNSCC tumorigenesis, but it has to be regarded as a favorable prognostic factor as far as response to treatment is concerned. Wood dust exposure surely is a risk factor for ITAC development, though there is no consensus on the mechanisms at work in this process. ITAC shares some similarities with colorectal cancer; however, it is not clear yet whether the same genetic mutations are involved in these two malignancies. Finally, ONB displays a huge variety of genetic abnormalities, so much that it is often subjected to subclassifications, but at the same time offering a number of targeted therapeutic options.

## Figures and Tables

**Figure 1 ijms-24-02670-f001:**
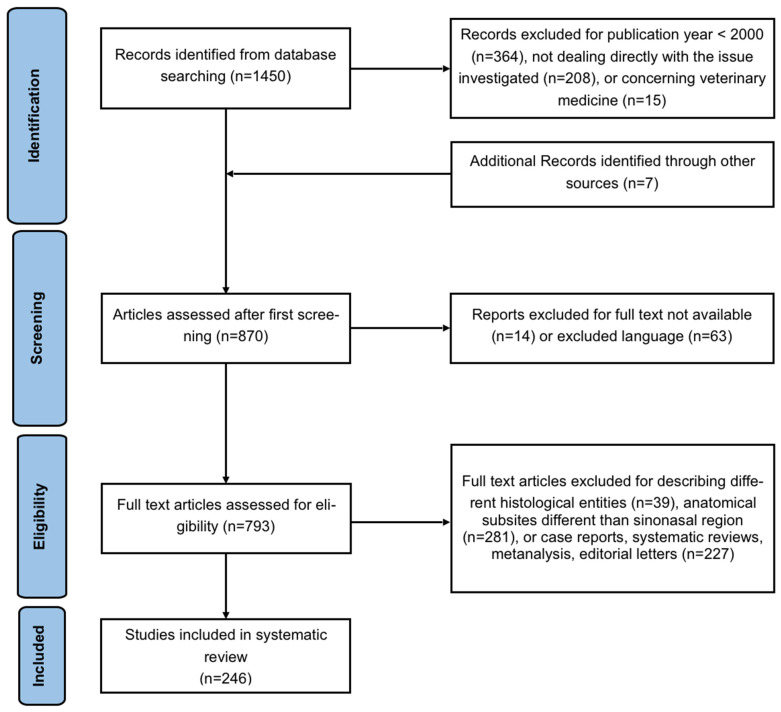
Flowchart of the article’s search and selection procedure according to the PRISMA criteria.

## Data Availability

The data presented in this study are openly available in online platforms (Pubmed, Scopus, and Medscape Ovid).

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
