# Peer review of "Tumors of the Nose and Paranasal Sinuses: Promoting Factors and Molecular Mechanisms—A Systematic Review"

_ijms, 2023, doi:10.3390/ijms24032670_

Round 1
Reviewer 1 Report
The present systematic is complete and well-written. Evidence about molecular features and mechanisms of sinonasal cancer are scarce, you performed a good summary of the state of art.
Author Response
Thank you for your kind review.
Reviewer 2 Report
This systematic review provides a wide summary of current knowledge about three of the most common histotypes of sinonasal malignant tumors. While other reviews on the molecular mechanisms of carcinogenesis in this category of tumors are already present in literature, this is to my knowledge the first systematic review conducted on this topic.
The review was well conducted according to PRISMA guidelines. While the search criteria fail to identify all the wide amount of available articles regarding the molecular mechanisms of the carcinogenesis of sinonasal tumors, the discussion nonetheless provides an interesting overview of the main oncogenetic pathways currently known in this type of tumors, even if the heterogeneity of the studies included inevitably allows only a qualitative synthesis of the data.
Some minor observations:
- Figure 1. should be edited since it is poorly formatted and some sentences are not entirely readable
- In Figure 2 the use of a pie chart could be misleading since the total number of article included is not 269 but 246 (I believe the difference is explained by some articles dealing with more than one histotype but this is not clearly stated)
- In table 1 on page 16 of 45, second line, 8th column, "Protein expression levels were higher in the IP and IP with SCC normal nasal mucosa" should be "Protein expression levels were higher in the IP and IP with SCC than in normal nasal mucosa"
Author Response
Thank you for your very kind review. We will edit Figures and Tables as per your suggestions.
Reviewer 3 Report
This systematic review article regarding sinonasal neoplasms promoting factors and molecular mechanisms is useful for the understanding the pathogenesis mechanism.
The possible influence of genetic aberrations and molecular mechanisms for each tumorous disease on therapeutic efficacy is needed to be discussed, as well.
For clarity the difference, the colors used in the chart in Figure 2 should be changed to vivid, like white, gray and black.
Author Response
Thank you for your kind review. Of course, as stated in the paper, the ultimate aim of a better understanding of the pathogenesis of sinonasal cancer is to broaden treatment options. However, the influence of genetic aberrations and molecular mechanisms on the efficacy of currently available treatments is beyond the scope of this manuscript. We will edit Figure 2 as per your suggestion. Thank you.